# Effect of Volatile Fatty Acids Accumulation on Biogas Production by Sludge-Feeding Thermophilic Anaerobic Digester and Predicting Process Parameters

**Intisar Nasser Al-Sulaimi** 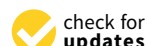, **Jagdeep Kumar Nayak, Halima Alhimali, Ahmed Sana** 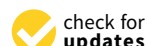 **and Abdullah Al-Mamun \*** 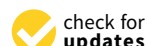



Department of Civil and Architectural Engineering, Sultan Qaboos University, Al-Khoud 123, Oman; intisar.sulaimi@owwsc.nama.om (I.N.A.-S.); j.kumar@squ.edu.om (J.K.N.); squ20092011@gmail.com (H.A.); sana@squ.edu.om (A.S.)
\* Correspondence: aalmamun@squ.edu.om

**Abstract:** Sewage sludge represents an important resource for reuse in the wastewater treatment field. Hence, thermophilic anaerobic digestion (TAD) could be an alternative technique to recover renewable resources from sludge. In the TAD biodegradation process, volatile fatty acids (VFAs) are the intermediate products of methanogenesis. However, the higher formation and accumulation of VFAs leads to microbial stress, resulting in acidification and failure of the digester. Therefore, several batch TADs have been investigated to evaluate the VFAs production from sludge and their impact on biogas generation and biodegradation efficiency. Three types of sewage sludges, e.g., primary sludge (PS), secondary sludge (SS), and mixed sludge (MS) were used as substrates to estimate the accumulation of VFAs and yield of methane gas. The system showed the maximum total VFAs accumulation from both PS and MS as $824.68 \pm 0.5$ mg/L and $236.67 \pm 0.5$ mg/L, respectively. The dominant VFA accumulation was identified as acetic acid, the main intermediate by-product of methane production. The produced biogas from PS and MS contained $66.75 \pm 0.5\%$ and $52.29 \pm 0.5\%$ methane, respectively. The high content of methane with PS-feeding digesters was due to the higher accumulation of VFAs (i.e., $824.68 \pm 0.5$ mg/L) in the TAD. The study also predicted the design parameters of TAD process by fitting the lab-scale experimental data with the well-known first-order kinetic and logistic models. Such predicted design parameters are significantly important before the large-scale application of the TAD process.

**Keywords:** thermophilic anaerobic digestion; volatile fatty acids; biogas production; process parameters

## 1. Introduction

The high quantity of sewage sludge from wastewater (WW) treatment poses a risk to the ecosystem as it contains around 50–80% of organic, toxic and perishable contaminants [1,2]. The quantity and the quality of the sludge also depend on the operating conditions and the characteristics of sewage. Therefore, the practice of dehydration and transportation of sludge for land applications such as biofertilizer or landfilling is not safe [3,4]. Moreover, the landfill of sewage sludge leads to hazardous greenhouse gas emissions and the leaching of toxins such as heavy metals and microplastics into the soil and groundwater [2,5–7]. On the contrary, sewage sludge is a burden to the municipal wastewater treatment plants (WWTPs). However, it can be an excellent source of value-added products that may offset the cost of WW treatment [8,9]. Therefore, more attention is needed to make sewage sludge management more sustainable and eco-friendlier [10,11].

Energy recovery from sewage Sludge is a vital step to modernize the conventional WWTPs. Besides energy recovery, they can be an efficient source of clean water and nutrients (i.e., ammonia and phosphorus) [8,12–15]. Such an energy/nutrients recovery from sewage sludge is sustainable and eco-friendly, which is a possible step for a circular

economy [16]. Anaerobic digestion (AD) is an eco-friendly biological process that efficiently converts the major fraction of wasted organics from food wastes, agricultural residues, animal manure, and sewage sludge to biogas (i.e., a mixture of $CH_4$, $H_2$, $CO_2$, $N_2$) and more stable sludges [17]. Therefore, it is considered as an essential process in modern WWTPs because of its economic and environmental benefits [18]. Furthermore, the recovered methane gas could be later converted to heat and electricity.

AD is a four stages biochemical process, e.g., hydrolysis, acidogenesis, acetogenesis and methanogenesis. In the hydrolysis and acidogenesis stages, the complex organic contents are converted into various short-chain volatile fatty acids (VFAs) such as acetic, propionic, lactic and butyric by acidogenic bacteria [14]. In the later stages, acetogenic microbes oxidized VFAs into acetate, hydrogen and carbon dioxide that are the main substrates for methane production by the methanogens [17]. Acetate is the dominating intermediate by-product for more than 75% of methane production in AD, while propionate and butyrate are other essential VFAs that will be converted into hydrogen and further acetate [19]. Enhancement of the biogas production in AD is dependent on various operational parameters such as the organic loading rate, C/N ratio, temperature, and hydraulic retention time [20,21]. Among the operational parameters, temperature played a major role in accelerating the digestion, methanogenesis and hydrolysis [22]. However, based on temperature requirements, the AD process is categorized into three types, i.e., (1) Psychrophilic (10–20 °C), (2) Mesophilic (30–38 °C), (3) Thermophilic (55–60 °C).

The higher temperature in the thermophilic AD (TAD) process achieved the faster degradation of organics, resulting in a quicker retention time and more methane production. In TAD process, the higher accumulation of VFAs as intermediate by-products acidified the digesters, causing the failure of the process [17]. However, limited studies investigated the effect of VFAs accumulation on the methanogenesis process. A recent study showed the inhibitory effect of acetic acid accumulation on biogas generation from kitchen waste [23]. Some other studies investigated the influence of other VFAs on biogas production from different substrates, i.e., food waste, water hyacinth, salvinia fruit processing wastewater and slaughterhouse waste [23–26]. A specific study observed a significant variation in VFAs accumulation when operated with secondary sludge (SS) as substrate. The study found that about half of the SS degraded easily, while the other half required additional pre/post-treatment, which was challenging for one-step SS digestion [27]. Furthermore, SS was more resistant than primary sludge (PS) due to lesser contents (only 20–30%) of mineralized organics in SS. So far, very limited studies were conducted to reveal the effect of total VFAs on biogas production using individual and combined sewage sludge in TAD. Therefore, it is essential to measure the optimum concentration of accumulated VFAs for the efficient performance of the TAD process.

Several mathematical simulations and modeling studies were done on the TAD process to predict the process parameters, where food wastes, agriculture waste, cattle slaughter wastewater, and manure were used as substrates [28–30]. Very limited information is available to predict the design parameters for TAD using the previously developed models for sewage sludges digestion. Therefore, the current study aimed to demonstrate the performance of a laboratory-scale TAD using the individual and combined sewage sludges (i.e., PS, SS and mixed sludge (MS)) as substrates to (i) observe the effect of VFAs accumulation on methane production and biodegradation, (ii) minimize the VFAs accumulation to optimize the process efficiency. The experimental data from the laboratory-scale demonstration were fitted by the well-known first-order kinetic and logistic models to predict the design parameters of TAD for large-scale implementation, when sewage sludge is used as digester feed.

## 2. Materials and Methods

### 2.1. Acclimation of Thermophilic Inoculum and Samples Preparation

PS and SS were collected from the primary and secondary clarifiers of WWTP at Oman wastewater service company, Muscat and stored at 4 °C to inhibit the further

digestion process. The characteristics of samples PS, SS and MS were measured and mentioned in Table 1 prior experiment. Initially, the mesophilic inoculum was allowed for growing at room temperature, and then the temperature was switched to thermophilic ranges (47 ± 0.2–57 ± 0.2 °C). After that, the system was kept for stabilization and the performance in terms of biogas production was monitored. Based on the literature and some preliminary studies, the acclimation of thermophilic inoculum can be considered within 11 days, which indicates the availability of thermophilic inoculum and the start-up of the process. Finally, the MS sample was prepared by mixing PS and SS with a 1:1 ratio.

**Table 1.** Characteristics of Primary Sludge, Secondary Sludge and Mixed Sludge (The results are average and standard deviation of 3 samples).

| | pH | COD (mg/L) | $BOD_5$ (mg/L) | SS (mg/L) | VSS (mg/L) | Partial Alkalinity mg/L as $CaCO_3$ | Total Alkalinity mg/L as $CaCO_3$ | $NH_3$-N Aqueous (mg/L) |
|---|---|---|---|---|---|---|---|---|
| **PS** | 7.72 ± 0.02 | 10400 ± 50.5 | 324 ± 45.5 | 11880 ± 100 | 10692 ± 110 | 342.86 ± 22.2 | 657.14 ± 23.5 | 50.85 ± 12.5 |
| **SS** | 7.76 ± 0.02 | 3368 ± 42.2 | 52.5 ± 65.3 | 2800 ± 150 | 2380 ± 120 | 28.57 ± 20.1 | 57.14 ± 12.5 | 70.24 ± 18 |
| **MS** | 7.54 ± 0.01 | 7648 ± 100 | 108 ± 50.2 | 6800 ± 105 | 6120 ± 125 | 64.29 ± 30.3 | 92.86 ± 20.1 | 11.98 ± 17.6 |

PS: Primary Sludge, MS: Mixed Sludge, SS: Secondary Sludge, COD: Chemical oxygen demand, $BOD_5$: 5 days biochemical oxygen demand, SS: Suspended solids, VSS: Volatile suspended solids.

## 2.2. Preparing of Reactor Configurations

The experiment was established in a lab-scale using 6 glass bioreactors with a 2 L working volume. Each anaerobic bioreactor was equipped with an overhead brushless DC motor for continuous mixing and a lid containing two ports (Catalog no 01-0102-01, Bioprocess instrument, Lund, Sweden). The motors were connected in series and driven by a motor controller (Catalog no 01-0109-02, Bioprocess instrument, Lund, Sweden). All the bioreactors were submerged in a hot water bath with an auto temperature and water level controller (Manufactured by Oman Electricity Distribution Company, Muscat, Oman). In the lid of the bioreactor, one port was used for feeding and sampling and the other port was used for gas collection. The gas outlet was connected to the measuring cylinders filled with water. The produced biogas was directly measured using the water displacement method as shown in Figure 1. An air-tight syringe transported the gas samples into the gas chromatograph (GC) for further analysis. Before the experiments began, the substrates and inoculum were kept at room temperature and deoxygenated by purging $N_2$ gas to maintain anaerobic conditions. The temperature of each reactor was maintained at 47 ± 0.2–57 ± 0.2 °C. Duplicate bioreactors were operated for each substrate condition and samples were tested in triplicate.

## 2.3. Analytical Methods

VFAs, such as, acetate, propionate, iso-butyrate and n-butyrate were analyzed with GC (Agilent HP brand, Santa Clara, CA, USA), with flame ionization detector column. The column temperature was maintained at 80 °C. The injector and detector were equipped with three carrier gases (i.e., air, Helium, Hydrogen) at a flow rate of 3.0 mL/min each simultaneously. Before injecting the digester effluents into the GC, they were centrifuged, filtered, and acidified. Then the purified samples were stored at 40 °C for not more than seven days before injecting into the GC.

The biogas compositions ($CH_4$, $CO_2$ and $N_2$) were also analyzed with GC, with a thermal conductive detector column at a temperature of 210 °C. Helium at a flow rate of 4.0 mL/min was used as the carrier gas. The main compounds detected were $CH_4$ and $CO_2$.

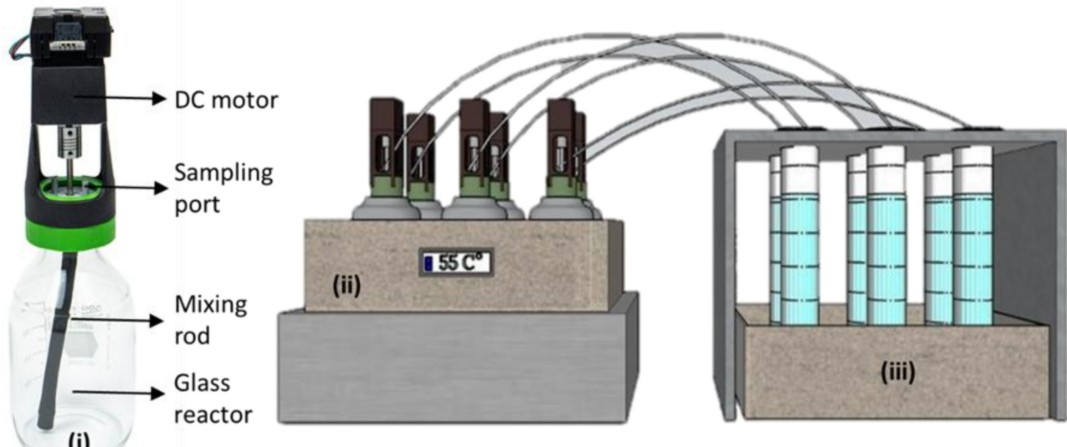

**Figure 1.** Experimental setup: (**i**) One anaerobic reactor with agitator, sampling and gas collection ports; (**ii**) Bioreactors submerged in a water bath; (**iii**) Water displacement cylinders for biogas collection and measuring.

Total suspended solids (TSS), volatile suspended solids (VSS), alkalinity and chemical oxygen demand (COD) concentrations were analyzed according to Standard Methods [31]. The alkalinity was estimated as partial alkalinity by titration to pH 5.75 and the total alkalinity by titration to pH 4.3. The measured alkalinity was calculated in mg/L as $CaCO_3$. Ammonium nitrogen ($NH_4^+$-N) was analyzed once in a week using the method reported by Björnsson et al. 2000 [32]. The volatile suspended solid (VSS) and biomass concentration were measured using Standard Methods. In addition, total solids and total volatile solids were obtained as the solid remaining after filtration and drying of the membrane at 105 °C.

The energy content (as electricity) from biogas was estimated using (Equation (1)).

$$E_{(ad)} = M \times CV_m \times EF_y \times (24/3600) \tag{1}$$

where, $E_{(ad)}$ is energy in kWh by anaerobic digestion, M is total methane production (kg), $CV_m$ is the calorific value of methane (55.5 MJ/kg), and $EF_y$ is the efficiency of typical gas engine or internal combustion engines, which is 30% [33].

*2.4. Kinetic Model Analysis*

Kinetic models were taken into consideration to describe the bacterial metabolic and substrate utilization process in TAD and evaluated methane yield from various organic substrates. Two kinetic models, i.e., a first-order (Equation (2)) and a logistic model (Equation (3)) were used to fit the cumulative methane generation to validate the experimental data. The models were selected from the literature to verify the TAD data as similar models were applied to describe the kinetics of methane production from AD processes [34–37]. All the process parameters of the applied kinetic and logistic models were determined and analyzed statistically using Microsoft Excel (version 2010).

The first-order kinetic model, Equation (2), was used to represent the biogas yield from bio-wastes using a conventional digestor.

$$P_t = P_m \times [1 - \exp(-kt)] \tag{2}$$

$$\text{Or } Ln(P_m - P_t) = -kt + LnP_m$$

where, $P_t$ is the cumulative biogas yield (mL $CH_4$/g VS) at time t, $P_m$ is the ultimate biogas production possible (mL $CH_4$/g VS), k is the first-order digestion rate constant (or hydrolysis constant) ($d^{-1}$), t is the digestion time (d). Using the experimental data, the linear graph of $Ln(P_m - P_t)$ versus t (simplified format of Equation (2)) estimated and validated the predicted k and $P_m$ values for the currently investigated TAD process.

Logistic model represented by Equation (3) showed the relationship between the rate of biogas production to the ultimate biogas production possible and the maximum production rate. The model also described the pattern of growth kinetics that was increased in the initial stage and finally reached a stable level [38].

$$P_t = \frac{P_m}{1 + \exp\left[4R_m\left(\frac{\lambda - t}{P_m}\right) + 2\right]} \tag{3}$$

$$Ln(P_m/P_t - 1) = (-4R_m/P_m) \times t + (4R_m/P_m.\lambda + 2)$$

where, $\lambda$ is the lag phase (d), $R_m$ is the maximum biogas production rate (mL/g VS/d). Using the experimental data, the linear graph of $Ln(P_m/P_t - 1)$ versus t (simplified format of Equation (3)) estimated and validated the predicted $R_m$ and $\lambda$ values for the currently investigated TAD process.

## 3. Results

### 3.1. Effect of VFAs Accumulation on Methane Production

The VFAs accumulation and conversion to biogas, their effect on methane production and the rate of total substrate biodegradation are the vital parameters to optimize the efficiency of the TAD. The accumulation of VFAs, such as, propionic acid, butyric acid, caproic acid, acetic acid, isovaleric acid and isobutyric acid were detected in the TAD when operated with PS, SS and MS as substrate (Figure 2). The observed trend of VFAs accumulation and their effect on methane production is illustrated in Figure 2. The types of VFAs produced and accumulated in the TAD with PS were diversified (i.e., acetic acid, isobutyric acid, isovaleric acid, caproic acid, butyric acid and propionic acid); while there were only four types with MS (i.e., acetic acid, isobutyric acid, isovaleric acid and caproic acid). The study also showed no VFAs production and accumulation with SS (data not shown). During PS and MS feeding digesters, the highest accumulation of total VFAs was observed as 824.68 mg/L and 236.67 mg/L on the 11th day of operation, respectively (Figure 2A,B). The highest acetic acid production was 445.57 mg/L with PS, while it was 184.23 mg/L with MS. The results confirmed that PS was potentially a good substrate to produce a higher amount of VFAs and acetic acid due to it's higher content of easily biodegradable, soluble, monomeric organics (i.e., glucose, fructose, amino acids, short-chain fatty acids). After day 11, the methanogens utilized the accumulated acetic acid to produce biogas until day 27 (Figure 2A). That is why the highest accumulation of biogas was noticed within days 27 to 32 (Figure 2C). The PS-feeding digestor also showed the maximum generation of other VFAs as isovaleric acid (195.76 mg/L) on the 13th day, propionic acid (171.70 mg/L) on the 13th day, caproic acid (144.36 mg/L) on the 20th day and isobutyric acid (104.53 mg/L) on 13th day. However, except for the acetic acid, none of the other VFAs were converted to methane gas by the methanogens, which is represented in Figure 2A. The cause of not converting the other VFAs to methane by methanogens was the complexity of the molecular structures of the other VFAs (either long-chain and branched-chain). The continuous increase of VFA production indicated the increase of microbial activity and the efficient growth of hydrolysis and acid-forming bacteria [39].

The rate of VFAs production with MS and SS was significantly low due to less content of simpler organics (i.e., glucose, fructose, amino acids, short-chain fatty acids). The major influence of acetic acid accumulation and conversion to methane gas from PS and MS is shown in Figure 2C,D. Methane production started to increase in the TAD reactor from the 8, when operated with both samples. The methane production increased from 5.40% (248.36 mL) to 66.75% (5667.35 mL) with PS and 3.34% (90.58 mL) to 52.29% (2865.70 mL) with MS. The maximum methane generation with PS was noticed on the 32nd day, whereas it was on the 27th day with MS. The absence of VFAs on the day of highest methane production indicated the complete conversion of VFAs to methane by methanogens. The lower production of methane with MS-feeding digestor might be due to less content of simpler and readily available organics in the MS sludge that was difficult to hydrolyze [40,41].

However, the SS-feeding digestor showed negligible methane production (data not shown). This was because the SS contained a very insignificant fraction of simpler and readily available organics. Moreover, it contained a higher fraction of the long-chain fatty acids and extracellular polymeric substances, which were resistant to biodegradation [42].

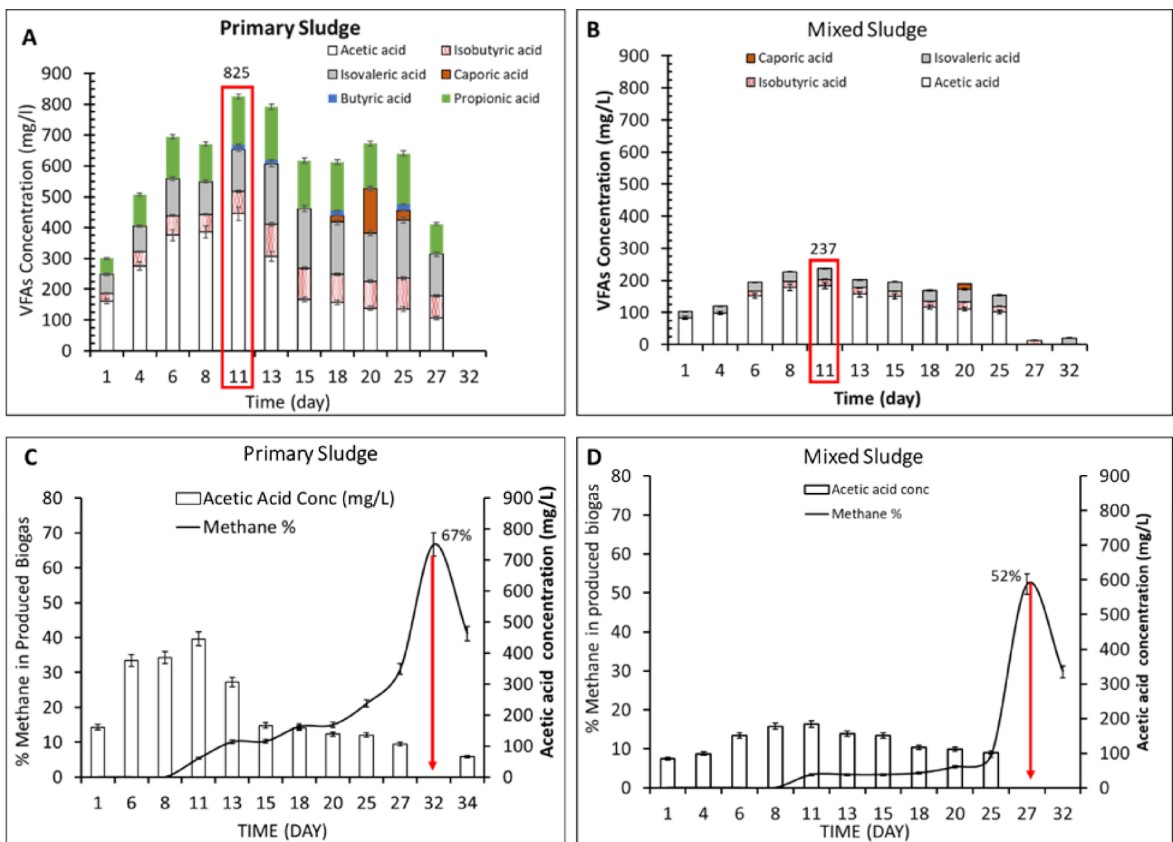

**Figure 2.** The effect of VFAs accumulation on methane production (%) in biogas for TAD process; (**A**) VFAs accumulations in TAD with primary sludge, (**B**) VFAs accumulations in TAD with mixed sludge, (**C**) Methane production (%) with primary sludge, (**D**) Methane production (%) with mixed sludge; Red square mark highlighted the day of maximum total VFAs and Red arrow mark showed the day of maximum methane.

### 3.2. Solids and COD Removal over the Operation

The removal of TSS, VSS, and COD is one of the contributing factors to the biogas production rate in TAD. The removal TSS, VSS and COD with the sludges is shown in Figure 3. The TSS contents in the influent of PS and MS were 11,880 mg/L and 6800 mg/L, respectively (Table 2). The maximum TSS removal with PS and MS was 66.33% and 58.09%, respectively (Figure 3A,C). In both cases, the TSS removal was approximately equal. Therefore, the results showed an insignificant effect of TSS removal on methane production. The influent VSS content in sludges were 10,692 mg/L and 6120 mg/L with PS and MS, respectively (Table 2). The PS-feeding digestor showed the highest removal of VSS as 7092 mg/L (65%) (Figure 3B). This was due to the higher fraction of simpler and readily available organics contents in the PS.

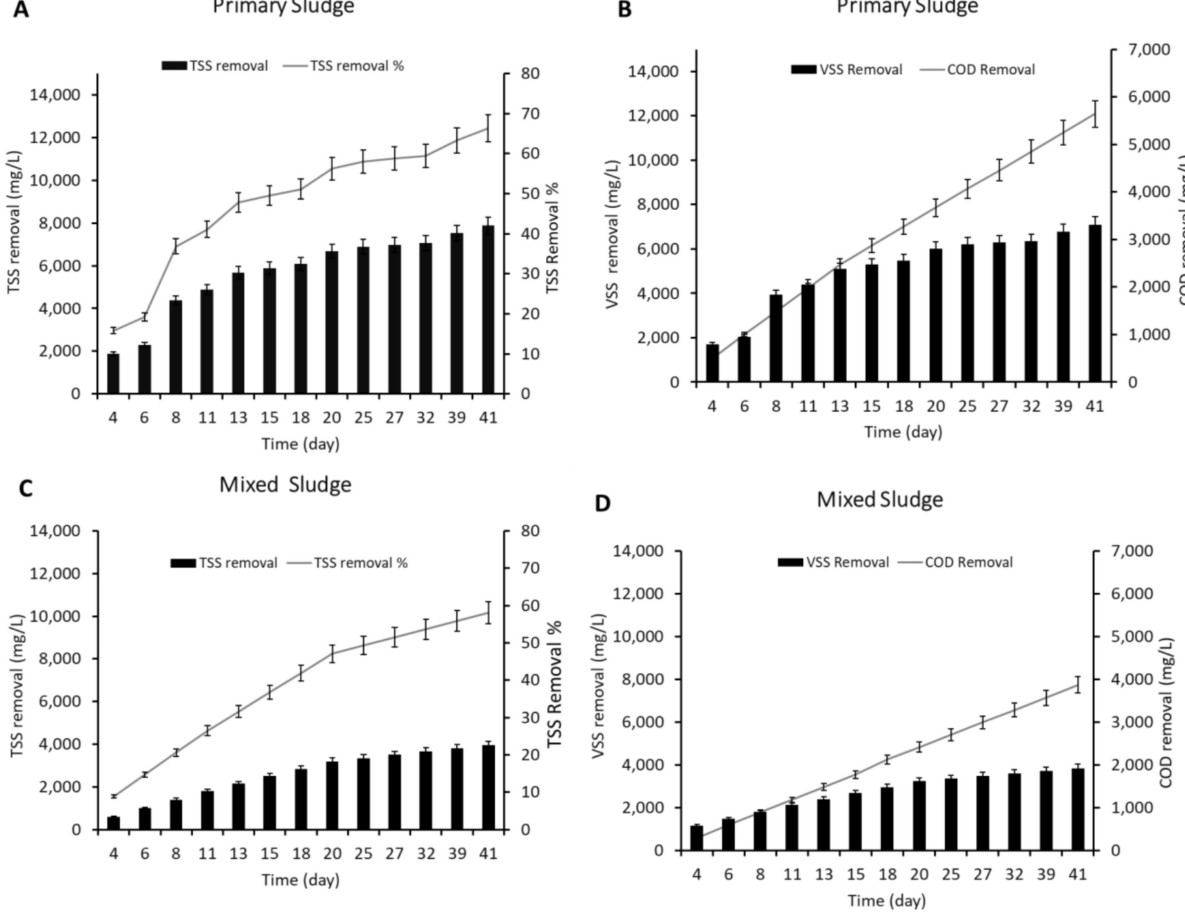

**Figure 3.** (**A**) and (**B**) represent TSS, VSS and COD removal from primary sludge; (**C**) and (**D**) represent TSS, VSS and COD removal from mixed sludge.

**Table 2.** Comparisons of methane production and power generation from various substrates in the literature.

| Configuration | Type of Feedstock | COD Removal (%) | Methane Production (% CH$_4$) | Power Generation (KWh) | Ref |
|---|---|---|---|---|---|
| Mesophilic AD | Mixed food waste | NA | 30 | 0.000927 | [10] |
| | Rice waste | | 64 | 0.002809 | [10] |
| | Date fruit waste | | <20 | 0.001224 | [10] |
| | Legume beans waste | | 40 | 0.001417 | [10] |
| UASB reactor with co-digestion | Sewage sludge and microalgal biomass pretreated by solar thermal system | 70 | 67.5 | - | [43] |
| UASB and AD combined with subsequent autotrophic nitrogen removal over nitrite process | Nitrogenous fertilizer Wastewater | 98.4 | - | - | [4] |

**Table 2.** *Cont.*

| Configuration | Type of Feedstock | COD Removal (%) | Methane Production (% CH$_4$) | Power Generation (KWh) | Ref |
|---|---|---|---|---|---|
| Acidogenic AD reactor | Beverage wastewater | 25 | 35 | - | [44] |
| Co-digestion fluidized-bed AD | Food waste and garden waste | - | 65 | - | [45] |
| Thermophilic AD | Sewage primary sludge | 54.21 | 67 | 0.00012 | This Study |

AD: Anaerobic digestion, USAB: Upflow sludge anaerobic bioreactor.

The ratio of VSS/TSS in the feeding substrate is another critical factor to monitor the microbial community activity and their diversification in the digestor. The experimental data showed the VSS/TSS ratios were 0.90, 0.85 and 0.80 for PS, SS and MS, respectively. The ratios for all the substrates used in the current TAD investigation were higher than those used in other ADs. Such a high VSS/TSS ratio indicated that most of the organics were in the format of VFAs, and therefore, a lesser amount of non-biodegradable organics was accumulated in the system [46].

The expected COD removal with all the substrates was noticed from the 1st day until the digesting finished due to endogenous biomass decay and methanogenesis (Figure 3B,D). The COD degradation rate was approximately constant during the entire cycle. Similar findings were also observed in several studies [47–49]. The study showed a maximum COD removal of 54.21% with PS, which was slightly lower than another reported study with alcohol industry wastewater (66.81%) [50]. However, the COD removal with MS was 50.6%. This result showed that the high COD removal with PS was directly proportional to high methane production and microbial biodegradation activities, as illustrated in Figure 4. Furthermore, the degradation of organics content was one of the primary mechanisms to accelerate methane production during the TAD process. The COD to VSS ratio of all the substrates was within the range of 1.29–1.72, which was almost similar to several reported studies in the literature [50,51].

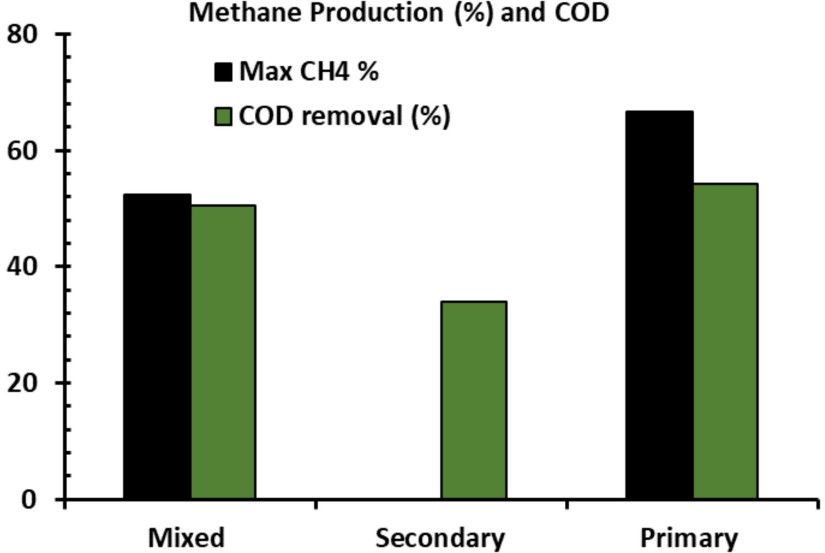

**Figure 4.** Relation of COD removal and the methane production in the different Samples.

### 3.3. Potential of Power Generation from Sludge as TAD Feedstock

The potential of TAD is majorly dependent upon the compositions of feedstock and their concentration. Many studies have been done with a large spectrum of feedstock, such as food waste, industrial sludge, rice waste, etc. [52]. The PS digester revealed

higher methane production over time than the MS digester and SS digester. However, the generation of methane was insignificant in the case of SS digester, which might be due to the less availability of simpler organics matter presented in the SS and the availability of aerobic bacteria.

In the context of energy recovery, sewage sludge could be used for the production of energy/biofuel either by anaerobic digestion, heat energy, co-incineration in coal power plants, gasification of biomass, wet oxidation, pyrolysis, high-temperature hydrolysis, ethanol acetone, etc., or direct electricity generation in a microbial fuel cell [8,15,53].

The current study determined recovered power generation and compared the value with previous studies in Table 2. The power generation value with SS was zero due to the negligible amount of methane generation. The maximum methane production rate as 1047.19 mL/g dry matter (DM) was noticed with a total biogas volume of 1570 mL/g DM comprised 67% occupancy (Figure 5). The calculated power generation of 0.12 Wh (0.00012 KWh) was seen, close to the power generation of mixed food waste, date fruit waste and legume bean waste [10] as indicated in Table 2. The process and the organic composition could be the reason for less methane production. The TAD operated with MS showed less methane due to the low organic content.

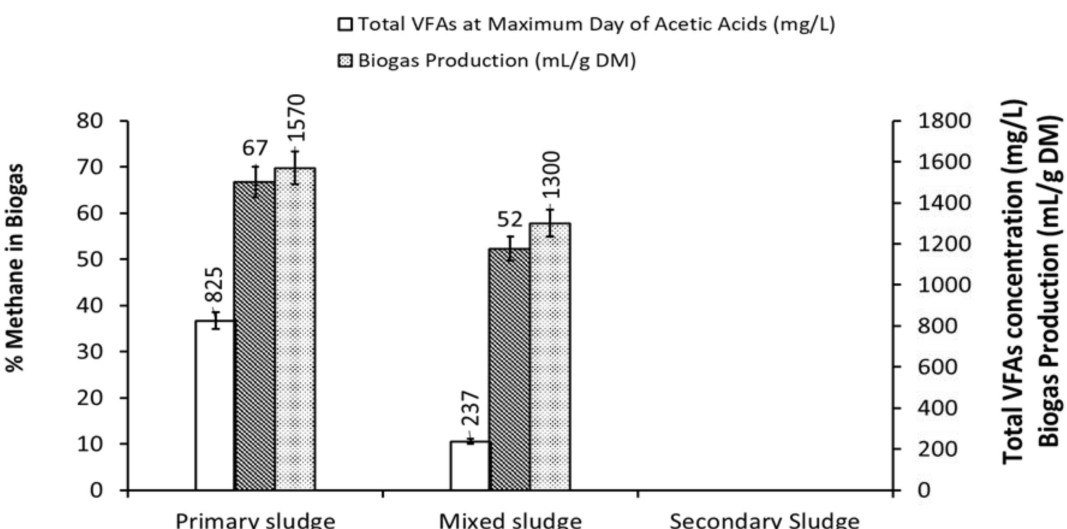

**Figure 5.** The maximum total VFAs with the maximum biogas production rate (mL/g DM).

### 3.4. Kinetic Models Study

The appropriate fitting of kinetic modeling is required to explain the mechanism and metabolic pathways associated with the TAD and its substrate utilization rate for different operational conditions. The models estimate the optimum design parameters, analyze the process efficiency and validate the experimental data for the current investigations [38,54]. Various mathematical models were available in the literature based on the process characteristics that represented the relationship between process parameters and process efficiency. For this study, two specific models, i.e., first-order kinetic (Equation (2)) and logistic (Equation (3)) models were applied to estimate the design parameters, e.g., the maximum biogas production possible ($P_m$, mL/g VS) and the first-order digestion rate ($k$, $d^{-1}$); lag phase ($\lambda$, d), the specific biogas production rate ($R_m$, mL/g VS/d) for process intensification and long-term operation in future.

The first-order kinetic model is the most widely used model when hydrolysis is the rate-limiting step in the AD process that predicts the ultimate biogas production. The logistic model was used to predict growth kinetics that increased in the initial stage leading to stabilization.

The experimental data of the cumulative biogas yield utilizing three different substrates, i.e., PS, SS, MS in TAD were represented in Figure 6, aiming to develop the mathematical equation for both models. The experimental data plot estimated the average trends, which ultimately formulated the predicted equation of the models. The predicted process parameters from the modeled equations were summarized in Table 3. The results showed that both the tested models fit the experimental data reasonably with significant determination coefficients ($R^2$) (Figure 6 and Table 3). The first-order digestion rate constant, k, for the three substrates varied from $4.7 \times 10^{-2}$–$6.4 \times 10^{-2}$, which was quite lower than similar studies [28,30]. The variation in k values for different substrates might be due to the composition, complexity and biodegradable fraction of the substrates. The consistent k values could be due to the biodegradability and higher nutrients concentration of the substrate that was favorable for methane production. For the first-order kinetic equation, the highest determination coefficient ($R^2 \approx 0.884$) was noticed in the case of MS substrate with the faster rate of digestion constant, $k \approx 0.064 \ d^{-1}$.

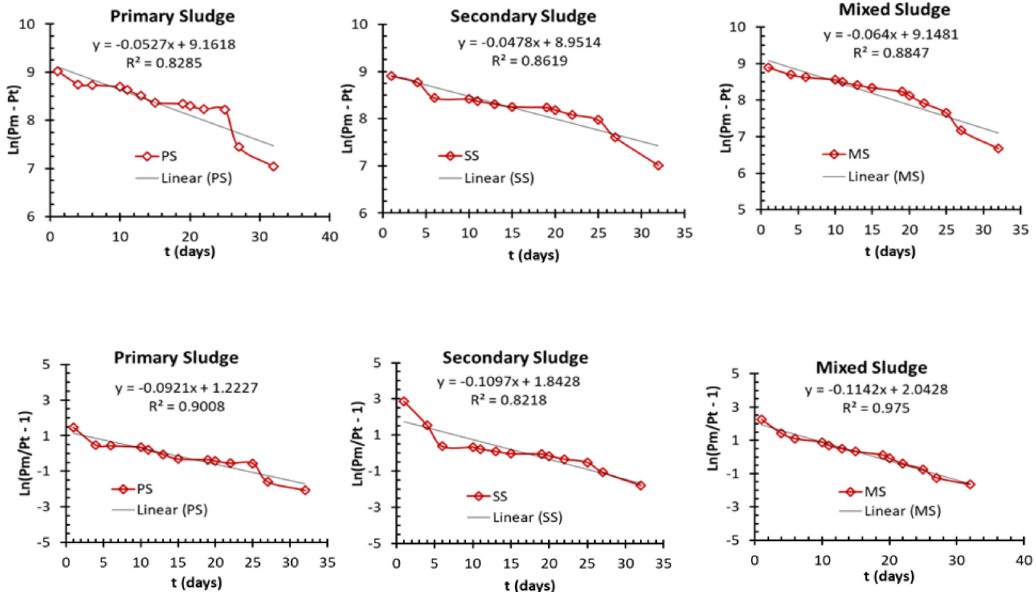

**Figure 6.** Predicting the design parameters from well-known First-order kinetic and Logistic models for three sewage sludge as substrates (PS, SS and MS) using the TAD reactor.

**Table 3.** Predicted design parameters from first-order kinetic and logistic models.

| Parameters | PS | SS | MS |
|---|---|---|---|
| Measured final biogas yield (mL biogas/g VS) for 100 days, $P_m$ | 10,200 | 7800 | 8080 |
| First-order kinetic model | | | |
| $P_m$ (mL biogas/g VS) | 9526 | 7718.6 | 9396.6 |
| k ($d^{-1}$) | 0.0527 | 0.0478 | 0.064 |
| $R^2$ | 0.829 | 0.862 | 0.884 |
| Error % for measured vs. predicted $P_m$ | 6 | 1 | 16 |
| Logistic model | | | |
| $P_m$ (mL biogas/g VS) | 9526 | 7718.6 | 9396.6 |
| $R_m$ (mL biogas/g VS/d) | 102 | 78 | 80.8 |
| $\lambda$ (d) | 8.4 | 1.29 | $\approx 0$ |
| $R^2$ | 0.901 | 0.821 | 0.975 |

Note: VS: Volatile solids.

The logistic model showed that the lag phase ($\lambda$, d) for MS substrate was approximately zero, which might be due to the readily available nutrients and enzymes concentration in the MS substrate. Such availability of nutrients and enzymes was favorable for microbial growth. However, there was a significantly high value of $\lambda$ observed with PS, which might be due to the low concentration of extracellular enzymes in PS substrate. Therefore, the predicted process parameters from the modeled equation are significantly useful to establish the large-scale application of the TAD system in the future.

### 4. Conclusions

The study observed the effect of VFAs accumulation on methane production and biodegradation efficiency for TAD process with three types of sewage sludges (i.e., PS, SS, MS). At every batch, when the total VFAs accumulation reached its maximum level (at day 11), the methane generation started and continued until day 32 at a significant rate. There was no trace of methane generation until the system reached its maximum accumulation of VFAs. Methane generation rates were 1047.19 mL/g DM with PS, while it was negligible with SS. The maximum VFAs accumulation with PS and MS were 824.68 ± 0.5 mg/L and 236.67 ± 0.5 mg/L, respectively. The acetogens and methanogens might transfer electrons between the microbial groups to promote the rapid conversion of acetic acid to methane. The precise effect of acetic acid on methane generations was noticed. The results indicated that TAD could be a better digester for sludge than other conventional treatments. The study also found the maximum COD removal at 54.21% and TSS removal at 66.33% with PS. However, no effect of TSS removal on methane production was observed. The experimental data were fitted with well-known first-order kinetic and logistic models to predict the design parameters for TAD. Therefore, the predicted process parameters will be useful for large-scale implementation of the TAD for recovering biogas from sewage sludge. More studies are required to speed up the methanogenesis process, explore the microbial community, and eradicate the pathogenic bacteria and viruses using TAD. In future research, a detailed exploration of microbial community analysis using next generation sequencing is required.

**Author Contributions:** Data curation, I.N.A.-S. and J.K.N.; Formal analysis, I.N.A.-S., J.K.N. and H.A.; Funding acquisition, A.S. and A.A.-M.; Investigation, H.A. and A.A.-M.; Methodology, I.N.A.-S. and H.A.; Project administration, I.N.A.-S., J.K.N., H.A. and A.S.; Resources, A.S. and A.A.-M.; Supervision, A.A.-M.; Writing—original draft, I.N.A.-S.; Writing—review & editing, J.K.N., A.S. and A.A.-M. All authors have read and agreed to the published version of the manuscript.

**Funding:** This research was funded by [The Research Council (TRC)] Fund Number [RC/RG-ENG/CAED/19/01]. And The APC was funded by the Fund (RC/RG-ENG/CAED/19/01), Oman.

**Institutional Review Board Statement:** Not applicable.

**Informed Consent Statement:** Not applicable.

**Acknowledgments:** The authors wish to extend their appreciation to The Research Council (TRC), Oman for the financial support through the Fund (RC/RG-ENG/CAED/19/01). The authors also appreciate Sultan Qaboos University and Oman Water and Wastewater Service Company, Muscat, Oman, providing all the logistic and technical support for the research.

**Conflicts of Interest:** All authors confirm that they have no conflict of interest to declare.

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
