# Peer review of "Effect of Volatile Fatty Acids Accumulation on Biogas Production by Sludge-Feeding Thermophilic Anaerobic Digester and Predicting Process Parameters"

_fermentation, doi:10.3390/fermentation8040184_

Round 1

Reviewer 1 Report

This study describes sewage treatment, biogas production, and energy recovery through sustainable and eco-friendly sewage sludge treatment. Based on the studies of previous studies, the purpose and validity of this study are well established. Overall, it is well-organized. However, some corrections are required to publish the manuscript.

  1. The abstract and introduction describe thermophilic treatment for removal of pathogenic bacteria. Microbiome analysis data using next gen sequencing etc. before and after thermophilic treatment are required. However, if it is difficult to obtain analysis data, the content of pathogenic bacteria should be removed from the abstract and introduction.

  1. How much energy is consumed by thermophilic treatment to treat pathogenic bacteria, and is this sustainable? What other cost-effective ways to get rid of pathogenic bacteria?

  1. In future research, detailed exploration of microbial community analysis using advanced next gen sequencing is required.

Author Response

Manuscript No: fermentation-1663589 MDPI

Manuscript Title: Effect of volatile fatty acids accumulation on biogas production by sludge-feeding Thermophilic Anaerobic Digester and predicting process parameters

Dear Editor and Reviewers,

The authors gratefully acknowledge the respected reviewers for the critical and comprehensive comments on the manuscript. We revised the paper thoroughly according to the editorial requirements as well as the reviewers’ comments. We hope that the changes will be acceptable by the editor and the reviewer. We hope that this revised paper is now suitable for publication in your prestigious journal.

Reviewer #1:

The authors studied the effect of VFA accumulation on the performance of anaerobic digestion using three different types of sludge. The topic of this this study is attractive to readers. But improvements are needed. I have attached my comments as follows. If the authors could properly address them or revise accordingly, I think it can be published.

On behalf of all the authors, I would like to express my heartfelt appreciations to the respected reviewer for reviewing the manuscript. All the point-by-point revision is mentioned below and marked in Red as per Track Changes in the manuscript. The entire manuscript has been proof-read by professional reader. Language has been correct. It has not shown in the manuscript due to clumsiness.

  1. The language of the entire manuscript needs to be improved. For example, the first sentence of the abstract, saying "Sewage sludge treatment and reuse have become a serious concern. I assume the authors meant that there are concerns in sewage sludge treatment, not sewage sludge treatment is a concern. More similar issues are found, and I recommend the authors to revise them.

Response: We are highly thankful to the reviewer.  The language has been improved in the revised manuscript by professional reader. The changes have not been shown due to clumsiness.

  1. The authors used methane percentage in the biogas as the metric for evaluating anaerobic digestion performance. Why not methane volume or production rate

Response: Thank you so much for the constructive comment. The volume metric methane production has been added in lines 229-230. The below table demonstrates the volume metric methane production:

Would there be a scenario that two digesters have the same biogas composition but very different biogas production volume?

Response: Thank you for the valuable comment. As per the above table the scenario of two digesters having the same biogas composition but very different biogas production volume was not observed.

  1. In line 202, the authors mentioned no VFA production or accumulation with secondary sludge. Is there no methane production either? If so, what's the point of including secondary sludge in this study but not discussing it?

Response: Thank you so much for the valuable comment. There was no production and accumulation of VFAs and negligible methane production with secondary sludge digester. We included secondary sludge to compare the digester with primary sludge and mixed sludge (mixed of PS with SS) as highlighted by yellow in the manuscript in lines 206, 224, 235 and 284.  

  1. For the kinetic models study, I would suggest the authors to mention the key information about this model in session 3.4, using very brief words. Otherwise, the readers might have to go back to the method session for the information.

Response: Thank you so much for the constructive comment. The kinetic model key information has been added in the manuscript in section 3.4 in lines 317- 320.

5.The kinetic models were constructed based on methane yield and digestion time and no information regarding VFA concentration was used (please correct me if my understanding is wrong). How are the models relevant to the works conducted previously in this study? And would the effect of VFA accumulation be reflected in the kinetic models? And move on to the large picture, there is already mature mathematical models like ADM1 and new models using machine learning algorithms. Why do the authors believe the information from first-order kinetic and logistic models can guide the design of anaerobic digestion?

Response: Thank you so much for the valuable comment. The kinetic model used to represent the methane yield which was directly produced due to the accumulation of VFAs. Sure, there are new models available but we focused in the basic first order kinetic and logistic mode in which we used the Logistic Model to get the lag phase where in first order kinetic model to get the hydrolysis constant.

Reviewer 2 Report

The authors studied the effect of VFA accumulation on the performance of anaerobic digestion using three different types of sludge. The topic of this this study is attractive to readers. But improvements are needed. I have attached my comments as follows. If the authors could properly address them or revise accordingly, I think it can be published.

  1. The language of the entire manuscript needs to be improved. For example, the first sentence of the abstract, saying "Sewage sludge treatment and reuse have become a serious concern. I assume the authors meant that there are concerns in sewage sludge treatment, not sewage sludge treatment is a concern. More similar issues are found and I recommend the authors to revise them.
  2. The authors used methane percentage in the biogas as the metric for evaluating anaerobic digestion performance. Why not methane volume or production rate? Would there be a scenario that two digesters have the same biogas composition but very different biogas production volume?
  3. In line 202, the authors mentioned no VFA production or accumulation with secondary sludge. Is there no methane production either? If so, what's the point of including secondary sludge in this study but not discussing it?
  4. For the kinetic models study, I would suggest the authors to mention the key information about this model in session 3.4, using very brief words. Otherwise, the readers might have to go back to the method session for the information.
  5. The kinetic models were constructed based on methane yield and digestion time and no information regarding VFA concentration was used (please correct me if my understanding is wrong). How are the models relevant to the works conducted previously in this study? And would the effect of VFA accumulation be reflected in the kinetic models? And move on to the large picture, there is already mature mathematical models like ADM1 and new models using machine learning algorithms. Why do the authors believe the information from first-order kinetic and logistic models can guide the design of anaerobic digestion?

Author Response

Manuscript No: fermentation-1663589 MDPI

Manuscript Title: Effect of volatile fatty acids accumulation on biogas production by sludge-feeding Thermophilic Anaerobic Digester and predicting process parameters

Dear Editor and Reviewers,

The authors gratefully acknowledge the respected reviewers for the critical and comprehensive comments on the manuscript. We revised the paper thoroughly according to the editorial requirements as well as the reviewers’ comments. We hope that the changes will be acceptable by the editor and the reviewer. We hope that this revised paper is now suitable for publication in your prestigious journal.

Reviewer #2:

This study describes sewage treatment, biogas production, and energy recovery through sustainable and eco-friendly sewage sludge treatment. Based on the studies of previous studies, the purpose and validity of this study are well established. Overall, it is well-organized. However, some corrections are required to publish the manuscript.

On behalf of all the authors, we would like to express our heartfelt appreciations to the respected reviewer for the remark time put into reviewing our manuscript.

  1. The abstract and introduction describe thermophilic treatment for removal of pathogenic bacteria. Microbiome analysis data using next gen sequencing etc. before and after thermophilic treatment are required. However, if it is difficult to obtain analysis data, the content of pathogenic bacteria should be removed from the abstract and introduction.

Response: Thank you so much for the constructive comment. The abstract and introduction sections content has been revised in lines 35-39, 54-56 and 74-78.

  1. How much energy is consumed by thermophilic treatment to treat pathogenic bacteria, and is this sustainable?

Response: Thank you so much for the valuable comment. The power consumption used for the thermophilic anaerobic digestion was not studied in this study. However, the author mention in section 3.3 the estimated potential energy generated from the produced methane in the biogas which can be used to heat the TAD as self-sustainable system and achieve the sustainability.  

What other cost-effective ways to get rid of pathogenic bacteria?

Response: Thank you so much for the valuable comment. There are some cost-effective ways to get rid of pathogenic bacteria such as potentially low-cost filter materials coated with silver nanoparticles were developed for the disinfection of groundwater [1]. In addition, Heterogeneous photocatalysis using titanium dioxide is a safe, nonhazardous, and ecofriendly process which does not produce any harmful byproducts. Extensive research in this field has been done in the area of photocatalytic removal of organic, inorganic, and microbial pollutants [2]. However, there are certain pathogenic (Helminths Eggs) that might find in the wastewater and sewage sludge. Helminth eggs represent an important challenge to environmental engineers as they are among the most difficult biological parasites to inactivate in wastewater and sludge [3]. Thus Thermophilic anaerobic digestion (TAD) is considered an advanced technology as a thermal treatment and could be an alternative method pathogen reduction. 

  1. In future research, detailed exploration of microbial community analysis using advanced next gen sequencing is required.

Response: Thank you so much for the constructive comment. The recommendation has been added in the revised manuscript in lines 365-366.

  1. Schoenen, D. Role of Disinfection in Suppressing the Spread of Pathogens with Drinking Water: Possibilities and Limitations. Water Res. 2002, 36, 3874–3888, doi:10.1016/S0043-1354(02)00076-3.
  2. Zhang, Z.; Gamage, J. Applications of Photocatalytic Disinfection. Int. J. Photoenergy 2010, 2010, doi:10.1155/2010/764870.
  3. Jiménez, B.; Maya, C.; Barrios, J.A.; Navarro, I. Helminths and Their Role in Environmental Engineering. Hum. Helminthiasis 2017, doi:10.5772/64878.

Round 2

Reviewer 2 Report

The revised manuscript addressed the comments from reviewer#2 properly. Now it's ok to be published.